# Effects of Infrared and Microwave Radiation on the Bioactive Compounds of Microalga *Spirulina platensis* after Continuous and Intermittent Drying

**DOI:** 10.3390/molecules28165963

**Published:** 2023-08-09

**Authors:** Neiton C. Silva, Isabelle S. Graton, Claudio R. Duarte, Marcos A. S. Barrozo

**Affiliations:** Faculty of Chemical Engineering, Federal University of Uberlândia, Block K, Campus Santa Mônica, Uberlandia 38400-902, MG, Brazil; neiton.silva@ufu.br (N.C.S.); bellegraton@hotmail.com (I.S.G.); claudioduarte@ufu.br (C.R.D.)

**Keywords:** *Spirulina platensis*, microwave, infrared, intermittent drying, bioactive compounds

## Abstract

Pharmaceutical, nutritional and food industries have recently become interested in the potential of *Spirulina platensis*, a kind of cyanobacterium with high levels of proteins, vitamins and bioactive compounds. Because of its high moisture, this microalga needs to be submitted to a preservation technique such as drying to be properly used. The aim of this work is to investigate the use of infrared and microwave radiation in the *Spirulina platensis* drying process. The experiments were performed in continuous and intermittent modes, evaluating different operating conditions for infrared and microwave drying, as well as their effects on the quality of the final product, expressed by the content of bioactive compounds (i.e., total phenolic, total flavonoid, citric acid and phycocyanin contents). The results proved that the use of electromagnetic radiation in the drying of spirulina is an interesting alternative for processing this material if performed under adequate operating conditions. The experiments carried out continuously at lower temperatures and powers and the combination between different temperatures and powers in the intermittent mode resulted in a final product with satisfactory levels of bioactive compounds and low operation times in comparison with conventional methodologies.

## 1. Introduction

*Spirulina platensis* is a species of cyanobacterium that has attracted the attention of researchers in different areas of knowledge due its qualities and nutraceutical properties. This microalga contains a high quantity of proteins, including all the essential amino acids, and small quantities of methionine, cystine and lysine. It has about 10 types of vitamins (e.g., A, E, K, B1, B2, B6 and B12) and minerals (e.g., potassium, iron, calcium, phosphorus, manganese, copper, zinc and magnesium). The presence of phenolic compounds such as caffeic, chlorogenic, salicylic, synaptic and trans-cinnamic acids in spirulina’s biomass can act individually or synergistically with other antifungal and antioxidant compounds, as previously reported [1,2,3]. In addition, the pigments present in spirulina cells include beta-carotene, chlorophyl and phycocyanin, the latter being an antioxidant and anti-inflammatory compound widely used in cosmetic and pharmaceutical industries [4,5,6,7,8].

Despite its vast potential, the biomass of fresh spirulina is highly perishable because of its elevated moisture content (which can exceed 90% on wet basis) and chemical composition, making it favorable to the proliferation of microorganisms. For this reason, the drying technique is an essential step in the effective use of this material [9,10,11,12]. Since this technique and the operating conditions can significantly affect the functional properties and nutritional value of this microalga, the correct choice of equipment and conditions can guarantee the quality of the product. Some studies have shown that conventional drying systems may be ineffective in preserving the nutritional quality of microalgae during the moisture removal process, in addition to adding high costs to the process if not performed under adequate conditions. Therefore, alternative methods that better preserve its functional and nutritional properties, especially the content of bioactive compounds, should be considered and evaluated [13,14,15].

In this context, the use of electromagnetic radiation (i.e., infrared and microwave) for moisture removal can be an interesting alternative for processing spirulina biomass due to its advantages over other conventional methods. Infrared is a type of radiation released by all bodies above 0 K (absolute zero temperature) whose frequency is immediately lower than that of the color red in the electromagnetic spectrum, hence the name “infrared”. In infrared drying, the energy penetrates the material surface and creates internal heating by molecular vibration, removing the moisture. The material is intensely heated, increasing the drying rate and saving energy. Among the positive points of infrared drying are its high efficiency in converting the supplied energy into heat, minimal heat loss to the surrounding and ease of control and handling [16,17,18,19].

The microwaves, on the other hand, are electromagnetic radiation with a high wavelength (from 1 mm to 1 m) and a frequency that varies between 300 MHz and 300 GHz. The main property of this radiation for the drying process is the agitation that the microwaves cause in water molecules within the materials resulting from dipolar and ionic mechanisms, producing volumetric and selective heating, and consequently leading to high heating rates and moisture removal with significantly reduced time and better uniformity. The microwave dryers are compact, easy to operate and energetically more efficient than conventional convective systems [20,21,22,23].

In addition to the use of radiative techniques, intermittent drying appears as a method that can decrease effective drying time and reduce energy consumption, improving the quality of the final product [24,25]. While under continuous drying the amount of energy supply throughout the process can result in quality degradation, heat damage to the surface of samples and wastage of heat energy—mainly in the last stages of the process when the drying rate decreases and the surface of samples becomes dry—under intermittent drying, the intensity and/or supply period of thermal energy to the process can be modified, thus allowing appropriate time for moisture transfer from the inner layers to the surface, minimizing heat damage [26,27,28].

There is limited information in the literature on the comparison between infrared and microwave drying in continuous and intermittent modes to process microalgae. These drying methodologies could become an interesting alternative for using all the potential of *Spirulina platensis* at the same time, which can possibly solve some problems observed in conventional drying techniques, such as degradation of bioactive compounds, high operational times and energy consumption. Therefore, the objective of the present study is to investigate the continuous and intermittent drying of the microalga *Spirulina platensis* using infrared and microwave radiation and the effects of their operating variables on the content of bioactive compounds, expressed by the total phenolic, total flavonoid, citric acid and phycocyanin contents.

## 2. Results and Discussion

### 2.1. Infrared Drying

#### 2.1.1. General Considerations

The visual appearance of spirulina samples dried using infrared radiation is presented in Figure 1. As observed, the material presents a polished and shiny surface, similar that observed by Desmorieux and Hernandez [29] in spirulina drying with different processes. Darkening regions can also be seen, especially in the experiments performed at higher temperatures. The dried microalga also showed high adherence to the plate support.

Figure 2 shows images of fresh spirulina and spirulina dried with infrared radiation, which were obtained using a scanning electron microscope (SEM). The fresh spirulina (Figure 2a) presented a series of filamentous structures (called trichomes), composed by cells in an elongated spiral form that is responsible for the nomenclature of this microalgae genus [5,6,30]. In all the samples dried using infrared radiation (Figure 2b–e), it was possible to observe that the moisture removal process significantly changed the material structure, which became more concise and uniform, due the fusion and agglomeration of the trichomes during drying. A similar result was verified by Desmorieux et al. [11] with SEM images of this microalga dried using convective drying.

The samples dried at 65 °C (Figure 2b) showed a uniform structure without pores or deformations. However, the presence of residual fusiform structures that increase in intensity as the temperature increases (Figure 2c, Figure 2d and Figure 2e, respectively, at 80 °C, 95 °C and 110 °C) may indicate that higher infrared exposure times change the material structure more than high temperatures (see drying times in Table 1 of the next subtopic).

#### 2.1.2. Final Moisture, Water Activity (a_w_) and Drying Kinetics

The final moisture, water activity (a_w_) and drying time obtained from the continuous infrared drying experiments are listed in Table 1.

The experiments performed at 50 °C did not result in adequate values of moisture and a_w_ (i.e., below 0.600) [31,32] at the end of the infrared drying process and were therefore discarded from the kinetic analysis. Costa et al. [7] observed similar behavior when using convective drying temperatures lower than 50 °C, which were considered insufficient for appropriate moisture removal since the driving force was reduced because of the lower levels of energy provided to the material. All experiments performed above 50 °C led to a dried material with low levels of moisture and adequate a_w_. Reduced drying times were observed in the experiments performed at higher temperatures.

Figure 3 shows the kinetic curves obtained for the infrared continuous drying experiments. The kinetic model that best represented the experimental data was that proposed by Midilli et al. [33], with a medium quadratic correlation coefficient (R^2^) of 0.9869 and estimated parameters, as presented in Table 2. As expected, the kinetic constant (k), which is related to the effective diffusivity of water through the material during drying [34], had its absolute value increased with increasing temperature.

#### 2.1.3. Bioactive Compounds

The total phenolic content (TPC), total flavonoid content (TFC), acidity (CA) and phycocyanin content (PC) found in fresh spirulina and after continuous and intermittent infrared drying of *Spirulina platensis* are shown in Figure 4. In this figure, the bioactive compounds’ contents are represented by bars with values in the left *y*-axis. The line curves represent the drying time observed in each experiment with their values shown in the right *y*-axis.

It can be noted that all analyzed compounds had a reduction in their values after infrared drying in comparison with the fresh material, indicating some sensitivity of spirulina to this electromagnetic radiation. The decrease in bioactive compound content was also observed in conventional spirulina drying [8,13] and can be explained by its sensitivity to heat exposure. The microalga does not have cellulose in its cellular wall like in plants, making it more vulnerable to the effects of heat during drying [5,35].

By analyzing the operating conditions applied during continuous drying, the best results for all compounds were found at the lowest infrared temperatures, indicating that the amount of energy used in the process has more influence than time (since in these conditions the drying times were higher, as shown in Table 1). Nevertheless, under specific conditions of intermittent drying, it was possible obtain a higher content of bioactive compounds than observed under continuous drying, suggesting that the combination of different temperatures has a positive effect, improving the results.

The presence of phenolic compounds reinforces the pharmacological properties of spirulina, that is, its anticarcinogenic, antimicrobial, anti-inflammatory and antitumoral activities [1,2]. According to Figure 4a, although the best result for total phenolic content (TPC) under continuous drying was obtained at 65 °C (319.22 mg gallic acid/100 g of sample in dry matter), which was about 69% of the TPC observed in fresh spirulina (462.12 mg gallic acid/100 g in dry matter), the intermittent combination of 65 → 50 °C increased the TPC by about 33%, reaching 423.67 mg gallic acid/100 g (dry matter), i.e., a value close to that observed in the fresh material. The combinations of 110 → 50 °C, 80 → 65 °C and 80 → 50 °C produced a TPC similar to that obtained in continuous drying at 80 °C and 95 °C. However, the use of intermittence led to higher drying times than those observed under continuous drying conditions.

Flavonoid compounds have important biological properties, such as antioxidant, anti-inflammatory, estrogenic and antimicrobial activities [1,36]. Nevertheless, their presence in microalgae has been poorly explored in the literature [15]. The total flavonoid content (TFC) observed in the infrared-dried spirulina can be seen in Figure 4b. As noted, these compounds were more sensitive to infrared radiation than the phenolic compounds. Even though under continuous drying the TFC of the fresh material (9.86 mg rutin/100 g of sample in dry matter) decreased by 64% under the best operating condition, i.e., a temperature of 65 °C, under intermittent drying, the TFC improved significantly for all temperature combinations evaluated. The combinations of 65 → 50 °C and 110 → 50 °C led to the best results for TFC, that is, 6.08 and 5.31 mg rutin/100 g in dry matter, respectively. It is noteworthy that the experiment performed at 110 → 50 °C resulted in a TFC about 50% higher than that obtained under the best continuous drying condition (at 65 °C), with a drying time approximately 25% lower (446 min).

The acidity (CA) is expressed by the content of citric acid, an important antioxidant compound and the most versatile acidulant used in the industry [37,38]. Figure 4c shows that the CA results achieved under continuous drying were similar to those obtained for the previous bioactive compounds (TPC and TFC), with the highest value being found in the sample dried at the lowest temperature (i.e., 65 °C), but at lower levels than observed for fresh microalga (5336.01 mg citric acid/100 g in dry matter). However, the use of intermittence did not have the same relevance as observed previously for TPC and TFC, since the best temperature combinations in the intermittent mode (110 → 50 °C) led to a CA value close to that observed in the best continuous drying condition (at 65 °C).

The phycocyanin present in spirulina is commercially used as a pigment in the food coloring and cosmetics industry. Nonetheless, several studies have demonstrated that this compound also has high antioxidant and anti-inflammatory properties, attracting the attention of researchers to its pharmaceutical and therapeutical applications [39,40,41]. The phycocyanin content (PC) found in fresh (14.55 g phycocyanin/100 g of sample in dry matter) and infrared-dried spirulina is displayed in Figure 4d. As noted, phycocyanin was the compound that showed the highest sensitivity to infrared radiation. In continuous drying at 65 °C, for example, the PC of the dried product was 5.81 g phycocyanin/100 g in dry matter, about 40% of the value found in the fresh microalga. Phycocyanin thermosensitivity was also observed in the literature using other drying methods, such as convective, heat pump, refractance window and rotary drum drying [9,12,15,42]. However, similar to what was observed for the phenolic and mainly flavonoid compounds, the use of intermittence proved to be positive, since in most temperature combinations the PC was higher than the value obtained during continuous drying. The best results were achieved in the combinations of 80 → 65 °C and 110 → 65 °C, reaching a PC of 6.63 and 7.17 g phycocyanin/100 g in dry matter, respectively, which is about 14% and 23% higher than that observed in continuous drying at 65 °C, but with drying times approximately 40% lower.

### 2.2. Microwave Drying

#### 2.2.1. General Considerations

The visual aspect of the samples dried using microwave radiation can be seen in Figure 5. The samples underwent significant volume reduction and intense darkening due to the high energy intensity of this kind of radiation, as previously reported [21,43,44]. Unlike what was observed during infrared drying, the microwave-dried samples were easily removed from the plate support, showing no adherence.

The images obtained using scanning electronic microscopy (SEM) are presented in Figure 6. Similar to what was observed under infrared drying (Figure 2), the microwave-dried samples showed a physical structure more uniform and concise than that observed in fresh spirulina biomass (Figure 6a). It was also possible to verify that the experiments performed under lower microwave powers (Figure 6b and Figure 6c, respectively, at 200 W and 280 W) showed a residual fusiform structure that practically disappeared in the higher powers (Figure 6d, Figure 6e and Figure 6f, respectively, at 480, 600 and 800 W), indicating that the intensity of the microwave power had more effect on the final structure of the material than the drying time (see drying times in Table 3 of the next subtopic).

#### 2.2.2. Final Moisture, Water Activity (a_w_) and Drying Kinetics

The final moisture, water activity (a_w_) and drying time obtained from the continuous microwave drying experiments are listed in Table 3. The drying process occurred much quicker than when using infrared drying (Table 1) or when using conventional methodologies [20,21]. Drying times between 11 and 70 min led to a material with an adequate a_w_ value [31,32] and low moisture content depending on the power used. At higher microwave powers (600 W and 800 W), the drying times were the shortest and closer. The drying time significantly increased with decreasing microwave power, going from 18 min at 480 W to 33 min at 280 W and 70 min to 200 W, in accordance with the results of infrared drying at lower temperatures.

The kinetic curves obtained during microwave drying are shown in Figure 7. The kinetic model that best represented the experimental data was that proposed by Midilli et al. [33], with a medium quadratic correlation coefficient (R^2^) of 0.9977 and estimated parameters as presented in Table 4. As expected, the kinetic constant (k) increased with increasing microwave power.

#### 2.2.3. Bioactive Compounds

The content of bioactive compounds of the product obtained from continuous and intermittent microwave drying experiments is displayed in Figure 8. The compounds also showed sensitivity to microwave radiation, having their values significantly reduced in comparison with the fresh material. By analyzing the continuous drying experiments, it is notable that the different microwave powers used did not have the same influence on the results as temperature during infrared drying. The bioactive compounds in the dried samples at higher microwave powers had similar contents than those obtained at lower powers.

The total phenolic content (TPC) of the samples (Figure 8a) dried in continuous mode was reduced by about 35% of the original content (462.12 mg gallic acid/100 g in dry matter), with no significant effect of the microwave power used. These TPC values were also lower than those obtained under continuous infrared drying (Figure 4a). The use of intermittence increased the TPC results in most operating conditions, mainly in the combinations that used lower powers, i.e., 600 → 280 W, 480 → 280 W, 280 → 200 W and 600 → 200 W, which produced a material with about 80% more phenolics than the content observed in the continuous mode. Even though the final drying time was higher in some experiments, the positive effects on TPC and the reduced time in most of the microwave drying experiments allow us to consider these experimental conditions for future application.

The effect of microwave radiation on the total flavonoid content (TFC) (Figure 8b) in continuous drying was lower than on phenolics and higher than that of infrared drying (Figure 4b), although they represent about 42% of the TFC of fresh spirulina (9.86 mg rutin/100 g of sample in dry matter). However, the use of intermittence did not efficiently increase such content. The best results were achieved in the combinations of 800 → 280 W and 480 → 280 W, which ended up increasing the TFC by about 30% and 16%, respectively, in relation to the value obtained in the best condition of continuous drying (at 200 W).

Figure 8c shows that the acidity (CA) of the samples dried using microwave radiation was considerably lower than that found in samples dried using infrared radiation (Figure 4c), indicating a higher sensitivity of citric acid to this kind of radiation. The CA obtained in the continuous drying experiments was between 20–30% of the content present in the fresh spirulina (5336.01 mg citric acid/100 g in dry matter). The best result was obtained from the experiment performed at the lowest microwave power, i.e., 200 W (1698.53 mg citric acid/100 g in dry matter). As observed under infrared drying, the use of intermittence was not considered relevant to improving the CA, since the best CA value was lower than that found under the continuous drying condition.

The phycocyanin content (PC) present in the microwave-dried samples (Figure 8d) showed different and interesting behavior compared to what was observed before under infrared drying (Figure 4d). Under continuous drying, the PC increased as a function of microwave power, indicating that microwave exposure time has a significant influence on this variable. The levels of PC found in the samples were also similar to those observed from infrared drying, reaching values of about 40% of the initial content in the microalga (14.55 g phycocyanin/100 g in dry matter). The use of intermittence did not significantly influence the PC results, with higher PC values being obtained in experiments performed in combinations of 800 → 200 W, 800 → 480 W and 600 → 400 W, which were closer to those found for the best continuous drying condition (at 800 W).

## 3. Materials and Methods

### 3.1. Raw Material

The microalga *Spirulina platensis* was provided by the Brasil Vital company, located in the Goiás State, in the Central-West region of Brazil. Prior to use, the material was stored in small portions in sealed polyethylene packages and frozen in a freezer at −18 °C until the beginning of the experiments.

### 3.2. Experimental Apparatus

The experiments were carried out using the experimental apparatus illustrated in Figure 9. The infrared dryer used (Gehaka IV 2500; São Paulo-SP, Brazil) consisted of an infrared emitter, a temperature sensor, a balance, a control panel where the moisture content removal was monitored, and metallic plates to accommodate the samples (Figure 9a). The microwave drying process was performed in a system comprising a microwave oven (NN-SF560WRU; Panasonic, Osaka, Japan, 800 W) coupled to an analytical balance (Shimadzu, Barueri, Brazil) to measure the weight of the samples during moisture removal (Figure 9b).

### 3.3. Experimental Design

The experimental design used herein is shown in Table 5. It was developed to evaluate the effects of the operating variables of continuous and intermittent infrared and microwave drying of *Spirulina platensis* on the content of bioactive compounds of the product obtained.

Under infrared dying, the continuous experiments were carried out at five distinct temperatures: 50, 65, 80, 95 and 110 °C. All five runs were performed until the moisture removal variation was lower than 0.1% per minute (criterion for stopping the experiments). The intermittent drying process was performed by combining the different temperatures used in continuous mode. The samples were initially submitted to higher temperatures until reaching a moisture content of about 50%, and then to lower temperatures until the moisture removal variation was lower than 0.1%. All combinations totaled 10 experiments, presented in Table 5. About 40 g of fresh spirulina was used in each experiment.

The microwave drying experiments were carried out in continuous mode at five distinct powers: 200, 280, 480, 600 and 800 W. All of the runs were performed until reaching a moisture range of 7.0–10.0%, controlled by mass variation on the balance. This criterion was used to avoid sample overexposure and carbonization regions, which are common in experiments with microwave radiation [20,43]. The intermittent drying process was first performed at higher powers until reaching a moisture content of about 50% (monitored by mass variation on the balance), and then at lower powers until the final moisture content was between 7.0–10.0%, totaling 10 experiments (Table 5). About 45 g of fresh spirulina was used in each experiment.

### 3.4. Moisture and Water Activity (aw) Analysis

The moisture content of the fresh and dried samples (wet basis) was determined using the oven method: 105 ± 3 °C for 24 h, based on the Association of Official Analytical Chemists (AOAC)’s official method [45]. Water activity (a_w_) was measured using a Novasina RS 232/TRD-200 instrument (Novasina, Zurich, Switzerland), a water activity device that measures the a_w_ using a temperature control system coupled to an infrared sensor that provides results with a precision of ± 0.001. The water activity is important to verify the physical, chemical and microbiological stability of dried materials. In general, the materials should be dried until reaching levels of water activity lower than 0.600, as under this condition, most bacteria, fungi and yeast have their activity and growing inhibited [31,32,46]. All measurements were performed in triplicate.

### 3.5. Scanning Electron Microscopy (SEM)

The fresh and dried spirulina biomasses were examined with the scanning electron microscopy (SEM) technique, using a Carl Zeiss microscope, EVO MA 10, to evaluate the microstructural characteristics of the material and the possible physical changes that may occur during dehydration [15,31]. The samples were attached to the microscope supports using a conductive carbon tape and then metallized with gold (Leica, SCD 050). A 10-kV acceleration voltage was used in the SEM analysis.

### 3.6. Drying Kinetics

In general, the dehydration kinetic equations are presented in the form of variation of a dimensionless moisture number (moisture ratio) as a function of time. The moisture ratio (MR) is given by Equation (1):(1)$$MR=\frac{M-M_{eq}}{M_{0}-M_{eq}},$$
where M is the moisture content at any time, $M_{eq}$ is the equilibrium moisture content, and $M_{0}$ is the initial moisture content.

A great number of empirical and semiempirical equations have been proposed in the literature to describe the drying kinetics of biological materials [47]. Table 6 presents the drying kinetics equations evaluated in this paper, where *k*, *n*, *A* and *B* are the model parameters. The kinetic model that best predicted the experimental results was selected based on a statistical analysis, considering the correlation coefficient (R^2^), the significance of parameters, and the distribution of residues [48].

### 3.7. Analysis of Bioactive Compounds

The fresh and dried samples had their bioactive compounds measured to investigate the impact of the different infrared and microwave drying conditions on the quality of the final product in addition to comparing the effects of the continuous and intermittent drying processes. All analyses were carried out in triplicate, and the content of bioactive compounds is expressed as mean value ± standard deviation. Student’s *t*-test at a 95% confidence level was applied to evaluate the difference in mean values. All statistical analyses were obtained using the Statistica v.12 software package (Statsoft).

#### 3.7.1. Total Phenolic Content (TPC)

The total phenolic content was determined with the Folin–Ciocalteau method [53], using gallic acid as a standard and performing spectrophotometric reading at 622 nm (V1200 spectrophotometer, VWR, Leuven, Belgium). The results are expressed in milligrams of gallic acid per 100 g of sample (dry matter).

#### 3.7.2. Total Flavonoid Content (TFC)

The total flavonoid content was quantified following the colorimetric method described by Zhishen et al. [54], using rutin as a standard and performing spectrophotometric reading at 450 nm (V1200 spectrophotometer, VWR, Leuven, Belgium). The results are expressed in milligrams of rutin per 100 g of sample (dry matter).

#### 3.7.3. Acidity or Citric Acid Content (CA)

The analysis of citric acid content in the samples, expressed by the total titratable acidity, was performed with titration using standardized 0.1 N NaOH, according the AOAC’s official method [45]. The results are expressed in milligrams of citric acid per 100 g of sample (dry matter).

#### 3.7.4. Phycocyanin Content (PC)

The extraction of phycocyanin was carried out based on the methodology reported by Costa et al. [7], using water as a solvent extractor and performing spectrophotometric reading at 620 nm and 652 nm (V1200 spectrophotometer, VWR, Leuven, Belgium). The phycocyanin content was obtained using Equation (2), as described by Bennett and Bogorad [55]:(2)$$PC=\frac{\left[{OD}_{620}-0.474\left({OD}_{652}\right)\right]}{5.34}$$
where PC is the phycocyanin content (mg/mL) and ${OD}_{620}$ and ${OD}_{652}$ are the optical densities of the samples at 620 nm and 652 nm, respectively. The results in all assays were subsequently converted into grams of phycocyanin per 100 g of sample (dry matter).

## 4. Conclusions

The use of infrared and microwave radiation proved to be an interesting alternative for the processing and conservation of the microalga *Spirulina platensis*. The experiments showed that it is possible to obtain a final product with satisfactory water activity (a_w_) and low moisture levels, extending the shelf life of this microalga through the minimization of microbial growth. The statistical discrimination study of the kinetic behavior revealed that the equation proposed by Midilli et al. [33] was the one that best represented the experimental data in both methodologies.

By analyzing the results obtained from the continuous infrared drying process, it was possible to observe that higher contents of bioactive compounds were achieved at lower temperatures, but at higher drying time. The use of intermittence through the combination of high and low temperatures led to similar or higher levels of bioactive compounds than those observed in continuous drying experiments, but at lower processing times.

Despite the low drying times obtained in microwave drying, the use of this radiation proved to be more harmful to this microalga than infrared drying, as it considerably reduced the content of bioactive compounds, especially the total phenolic and citric acid (acidity) content. The use of different microwave powers in continuous drying did not significantly affect the quality of the final product in comparison with the effect of temperature in infrared drying, since the contents of bioactive compounds were close to each other, independent of the power used. While the phycocyanin content slightly increased with increasing microwave power, the other bioactive compounds analyzed showed better results at lower powers. The use of intermittence increased some results of total phenolics and total flavonoid but did not influence the citric acid and phycocyanin contents, contrary to what was observed under infrared drying.

Based on the abovementioned results, it can be concluded that the use of infrared and microwave radiation to process the microalga *Spirulina platensis*, mainly in intermittent mode, is a good strategy for the efficient conservation of this microalga.

## Figures and Tables

**Figure 1 molecules-28-05963-f001:**
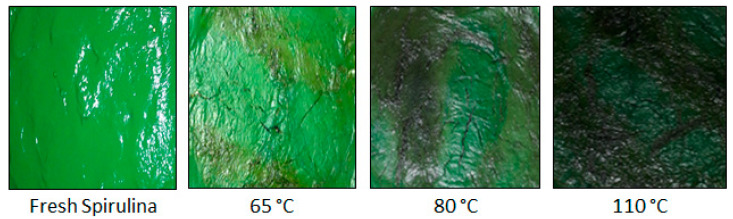
Visual representation of *Spirulina platensis* dried using infrared radiation.

**Figure 2 molecules-28-05963-f002:**
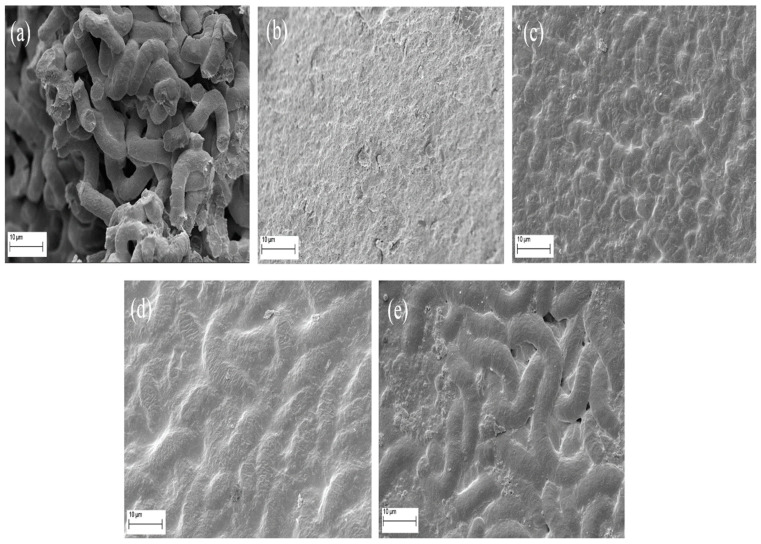
Scanning electron microscopy (SEM) images of *Spirulina platensis* dried using infrared radiation. Magnification of 4000 times for (**a**) fresh spirulina and spirulina dried at (**b**) 65 °C, (**c**) 80 °C, (**d**) 95 °C and (**e**) 110 °C.

**Figure 3 molecules-28-05963-f003:**
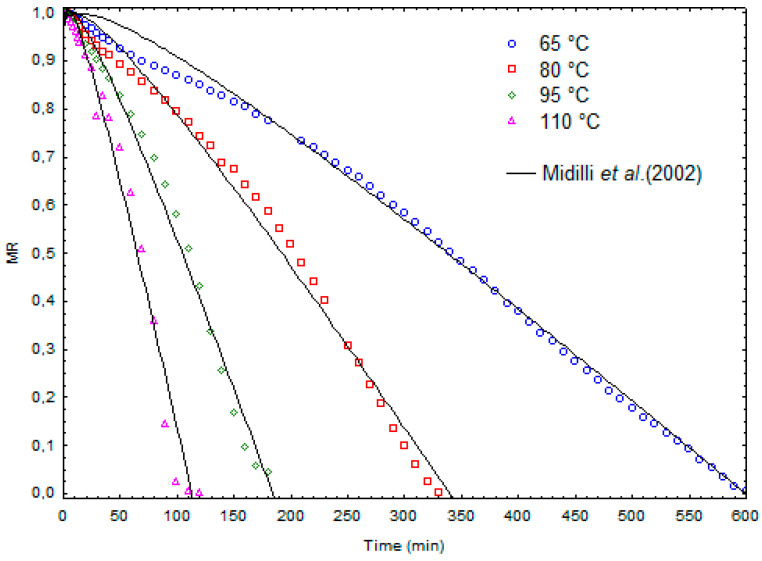
Infrared drying kinetics: experimental results and prediction according to the kinetic model proposed by Midilli et al. [33].

**Figure 4 molecules-28-05963-f004:**
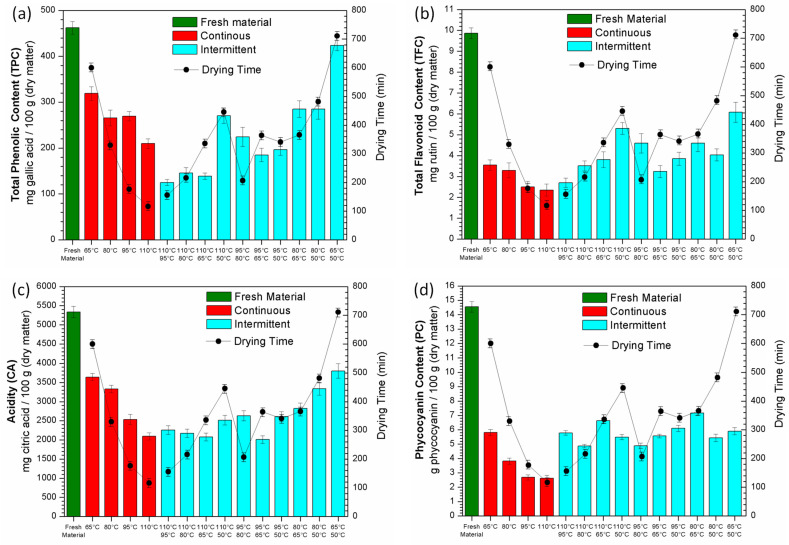
Content of bioactive compounds after infrared drying of *Spirulina platensis*: total phenolic content (TPC) (**a**), total flavonoid content (TFC) (**b**), acidity (CA) (**c**) and phycocyanin content (**d**).

**Figure 5 molecules-28-05963-f005:**
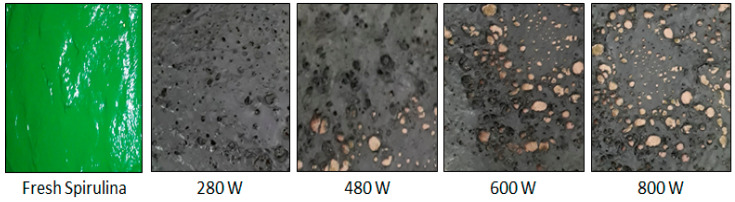
Visual representation of *Spirulina platensis* dried using microwave radiation.

**Figure 6 molecules-28-05963-f006:**
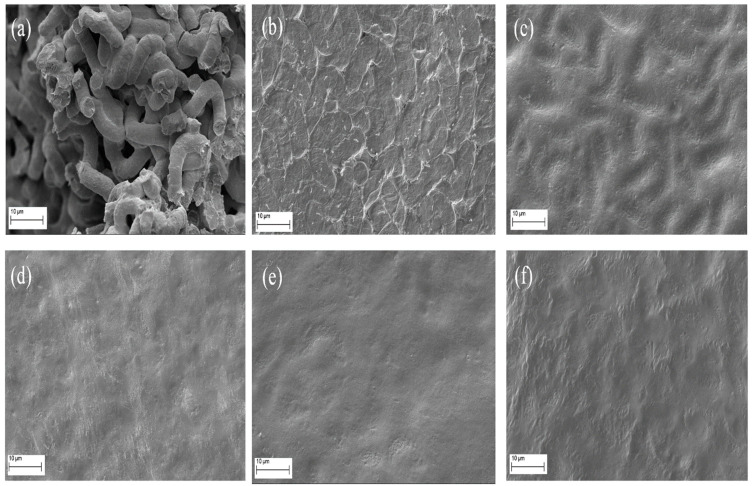
Scanning electron microscope (SEM) images of *Spirulina platensis* dried using microwave radiation. Magnification of 4000 times for (**a**) fresh spirulina and spirulina dried at (**b**) 200 W, (**c**) 280 W, (**d**) 480 W, (**e**) 600 W and (**f**) 800 W.

**Figure 7 molecules-28-05963-f007:**
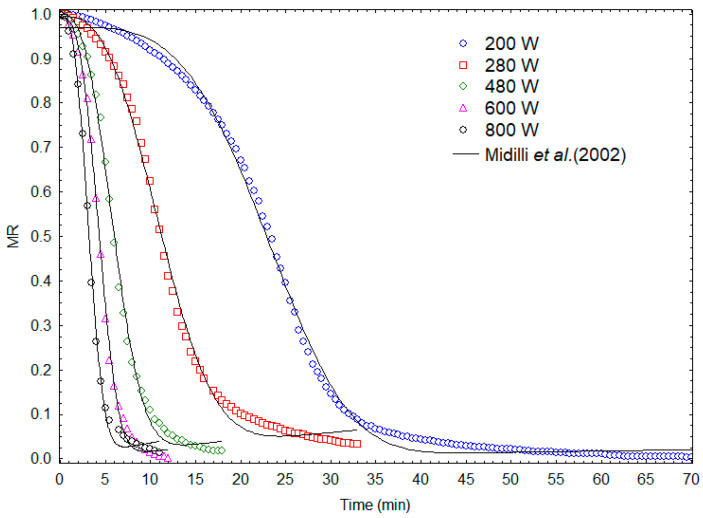
Microwave drying kinetics: experimental results and prediction according to the kinetic model proposed by Midilli et al. [33].

**Figure 8 molecules-28-05963-f008:**
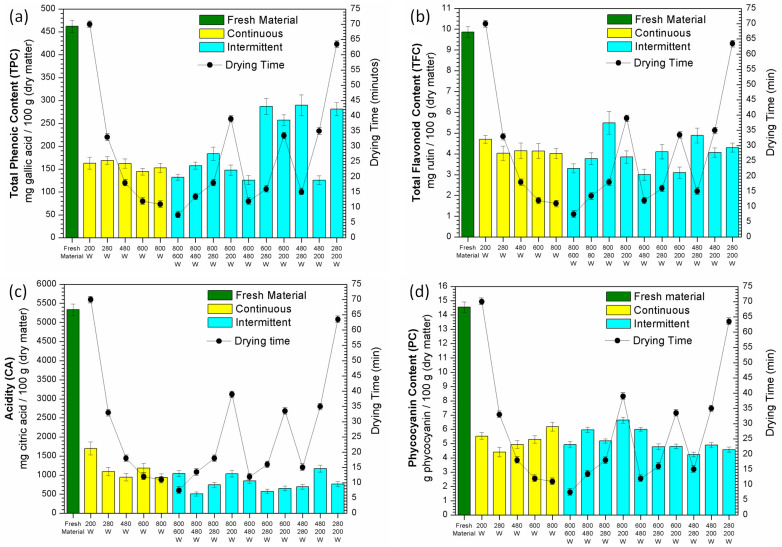
Content of bioactive compounds after microwave drying of *Spirulina platensis*: total phenolic content (TPC) (**a**), total flavonoid content (TFC) (**b**), acidity (CA) (**c**) and phycocyanin content (**d**).

**Figure 9 molecules-28-05963-f009:**
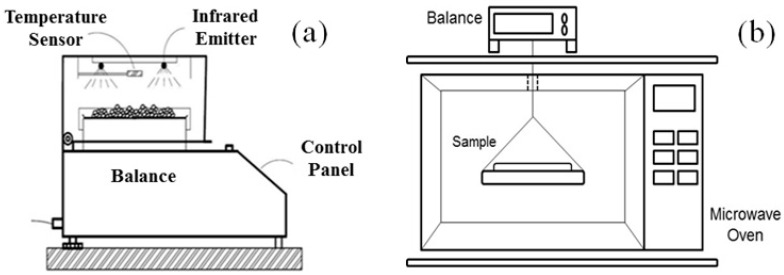
Drying experimental apparatus: schematic figures of infrared (**a**) and microwave dryers (**b**).

**Table 1 molecules-28-05963-t001:** Final moisture, water activity (a_w_) and drying time of continuous infrared drying experiments.

Temperature (°C)	Final Moisture (%)	Water Activity (a_w_)	Drying Time (min)
50 °C	28.72 ± 0.15%	0.736	720.0
65 °C	12.92 ± 0.46%	0.503	600.0
80 °C	4.97 ± 0.39%	0.444	330.0
95 °C	4.38 ± 0.31%	0.415	177.0
110 °C	4.25 ± 0.08%	0.348	117.0

**Table 2 molecules-28-05963-t002:** Kinetic parameters of infrared-dried spirulina following the model proposed by Midilli et al. [33].

Experiments	k	n	A	B	R^2^
65 °C	−1.62 × 10^−2^	0.54	0.9687	−2.72 × 10^−3^	0.9817
80 °C	−2.26 × 10^−2^	0.53	0.9689	−4.72 × 10^−3^	0.9918
95 °C	−3.29 × 10^−2^	0.53	0.9713	−8.98 × 10^−3^	0.9917
110 °C	−4.69 × 10^−2^	0.53	0.9711	−1.53 × 10^−2^	0.9825
				R^2^ medium	0.9869

**Table 3 molecules-28-05963-t003:** Final moisture, water activity (a_w_) and drying time of continuous microwave drying experiments.

Power (W)	Final Moisture (%)	Water Activity (aw)	Drying Time (min)
200 W	8.11 ± 0.03%	0.429	70.0
280 W	6.62 ± 0.26%	0.392	33.0
480 W	4.46 ± 0.19%	0.347	18.0
600 W	4.10 ± 0.08%	0.337	12.0
800 W	4.48 ± 0.23%	0.342	11.0

**Table 4 molecules-28-05963-t004:** Kinetic parameters of microwave-dried spirulina following the model proposed by Midilli et al. [33].

Experiments	k	n	A	B	R^2^
200 W	7.00 × 10^−6^	3.67	0.9698	2.93 × 10^−4^	0.9980
280 W	1.08 × 10^−3^	2.69	0.9935	1.97 × 10^−3^	0.9969
480 W	6.14 × 10^−3^	2.66	0.9998	2.19 × 10^−3^	0.9975
600 W	8.66 × 10^−3^	2.99	0.9905	1.70 × 10^−3^	0.9982
				R^2^ medium	0.9977

**Table 5 molecules-28-05963-t005:** Experimental design of infrared and microwave continuous and intermittent drying.

Drying	Experiment	Infrared	Microwave
Continuous	-	50, 65, 80, 95 and 110 °C	200, 280, 480, 600 and 800 W
Intermittent	1	110 → 95 °C	800 → 600 W
	2	110 → 80 °C	800 → 480 W
	3	110 → 65 °C	800 → 280 W
	4	110 → 50 °C	800 → 200 W
	5	95 → 80 °C	600 → 480 W
	6	95 → 65 °C	600 → 280 W
	7	95 → 50 °C	600 → 200 W
	8	80 → 65 °C	480 → 280 W
	9	80 → 50 °C	480 → 200 W
	10	65 → 50 °C	280 → 200 W

**Table 6 molecules-28-05963-t006:** Drying kinetic models.

Equation in the Literature	Reference
MR=exp⁡(−kt)	Lewis [49]
$MR=exp(-kt^{n})$	Page [50]
$MR=exp[-{\left(kt\right)}^{n}]$	Overhults et al. [51]
MR=Aexp⁡(−kt)	Brooker et al. [52]
$MR=A\cdot \exp\left(-kt^{n}\right)+Bt$	Midilli et al. [33]

## Data Availability

Data supporting reported results are available from the authors.

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
