# Peer review of "Effects of Infrared and Microwave Radiation on the Bioactive Compounds of Microalga *Spirulina platensis* after Continuous and Intermittent Drying"

_molecules, 2023, doi:10.3390/molecules28165963_

Round 1
Reviewer 1 Report
In general, the paper is well-written. It doesn't have major issues, and the conclusions are supported by the results and discussions. However, I believe that the SEM images could provide more information. Additionally, Figure 2b appears to be showing different information compared to the other images in Figure 2. It seems like it was taken in an area where there were no microorganisms. Would it be possible to replace it with another image that provides more information? Similarly, only in Figures 6a and 6b can the structures or shapes of the microorganisms be seen, but they are not visible in the other figures. I think it is necessary to provide more explanations regarding this matter.
Author Response
Below we reproduce the reviewers’ original comments, followed by our responses (R) in italics. Changes in the text in response to the reviewers have been highlighted in revised manusript.
Reviewer #1:
Comments and suggestions for authors:
In general, the paper is well-written. It doesn't have major issues, and the conclusions are supported by the results and discussions.
- However, I believe that the SEM images could provide more information.
R: New information about the SEM images (Figures 2 and 6) were now included in the text (new pages 3, 4 and 7).
- Additionally, Figure 2b appears to be showing different information compared to the other images in Figure 2. It seems like it was taken in an area where there were no microorganisms. Would it be possible to replace it with another image that provides more information?
R: Indeed, the samples dried by infrared radiation at 65 oC (Figure 2b) significantly changed the material structure which became more concise and uniform, because of the high drying time (600 min). New comments were included in the revised manuscript (new page 4).
3. Similarly, only in Figures 6a and 6b can the structures or shapes of the microorganisms be seen, but they are not visible in the other figures. I think it is necessary to provide more explanations regarding this matter.
R: In the experiments performed using microwave radiation we can observe that the samples showed a residual fusiform structure that practically disappeared in the higher powers, indicating that the intensity of the microwave power had higher effect in the final structure of the material than the drying time. New comments were included in the revised manuscript (new page 7).
The authors are grateful for the reviewer's valuable corrections that improved the manuscript.
Reviewer 2 Report
This work presents an interesting exploration and scientifically sound findings. However, it appears that the novelty of the study is not clearly established within the broader context of existing literature. Therefore, I suggest major revisions to enhance both the clarity and uniqueness of your contribution.
Please clarify and demonstrate the novelty of your study in relation to existing literature in the field. This would help to position your work within the broader research context and make a case for its distinctiveness.
In the section where you state, "Some studies have shown that conventional drying systems may be ineffective in preserving the quality of microalgae, besides adding high costs to the process if not performed under adequate conditions", further clarity is needed. Is the ineffectiveness due to a decrease in nutritional values or difficulties in effective water removal?
In your statement, "In all images it is possible to observe that the infrared exposure considerably changed the material structure, which became more uniform and with a solid leaf aspect", the term "leaf aspect" is unclear. Furthermore, the reason for the complete breakdown of the cellular structure at a low temperature (65ºC) needs to be explained.
An explanation is required for considering an aw value below 0.6 as adequate.
The claim that "lower temperatures can considerably increase the drying times, hindering the process and increasing energy consumption" needs evidence to support it. The usage of the term 'hinder' seems unfounded, and the statement about increased energy consumption is speculative without a corresponding energy balance.
The caption in Figure 4 is incomplete and the presence of two x-axes is confusing. Please rectify these issues.
In Figure 4, it is unclear why all values decrease at a single point (95-80). If drying time was replicated, error bars should be included. Additionally, it is essential to conduct statistical analyses to confirm significant differences between treatments.
There is a lack of discussion in your study. I recommend splitting the "Results and Discussion" section into two distinct parts to ensure a comprehensive presentation and discussion of your findings.
Finally, the entire manuscript should be presented in the English language to ensure universal readability and understanding.
I am confident that addressing these issues will significantly enhance your manuscript and better illustrate its contribution to the field.
The manuscript contains several grammatical and writing errors which detract from its overall quality. For instance, the sentence, "The aspect of Spirulina samples dried by infrared radiation is shown in Figure 1", needs revision.
Author Response
Below we reproduce the reviewers’ original comments, followed by our responses (R) in italics. Changes in the text in response to the reviewers have been highlighted in revised manusript.
Reviewer #2:
Comments and suggestions for authors:
This work presents an interesting exploration and scientifically sound findings. However, it appears that the novelty of the study is not clearly established within the broader context of existing literature. Therefore, I suggest major revisions to enhance both the clarity and uniqueness of your contribution.
- Please clarify and demonstrate the novelty of your study in relation to existing literature in the field. This would help to position your work within the broader research context and make a case for its distinctiveness.
R: New comments about the novelty of this study were included in the revised version of the manuscript (new page 2).
- In the section where you state, "Some studies have shown that conventional drying systems may be ineffective in preserving the quality of microalgae, besides adding high costs to the process if not performed under adequate conditions", further clarity is needed. Is the ineffectiveness due to a decrease in nutritional values or difficulties in effective water removal?
R: The ineffectiveness is due the decrease of the nutritional values expressed by the content of bioactive compounds. We included this information in the revised version of the manuscript (new page 2).
- In your statement, "In all images it is possible to observe that the infrared exposure considerably changed the material structure, which became more uniform and with a solid leaf aspect", the term "leaf aspect" is unclear. Furthermore, the reason for the complete breakdown of the cellular structure at a low temperature (65ºC) needs to be explained.
R: We agree with the reviewer. The term “leaf aspect” was removed in the revised version of the manuscript and more information about the SEM images were included in the text (new pages 3 and 4).
- An explanation is required for considering an aw value below 0.6 as adequate.
R: The water activity is important to verify the physical, chemical and microbiological stability of dried materials. In general, the materials should be dried until reaching levels of water activity lower than 0.60, because in this condition most bacteria, fungi and yeast have their activity and growing inhibited, as related by Chen and Patel [29] and Sablani and Rahman [30]. These information are included in the Section 3.4 of the Material and Methods sections (new page 12).
- The claim that "lower temperatures can considerably increase the drying times, hindering the process and increasing energy consumption" needs evidence to support it. The usage of the term 'hinder' seems unfounded, and the statement about increased energy consumption is speculative without a corresponding energy balance.
R: We agree with the reviewer observations. This part of the text was modified in the revised version of the manuscript (new page 4).
- The caption in Figure 4 is incomplete and the presence of two x-axes is confusing. Please rectify these issues.
R: In the Figures 4 and 8, the bioactive compounds contents are represented by the bars with their values in the left y-axis. The line curves represent the drying times observed in the experiments with their values showed in the right y-axis. This explanation was added in the revised version of the manuscript (new page 6).
- In Figure 4, it is unclear why all values decrease at a single point (95-80). If drying time was replicated, error bars should be included. Additionally, it is essential to conduct statistical analyses to confirm significant differences between treatments.
R: In infrared drying the intermittent process was performed by combining the different temperatures used in the continuous mode. The samples were initially submitted to higher temperatures until reaching a moisture content of about 50%, and then to lower temperatures until the moisture removal variation was lower than 0.1 (new page 12). In the specific condition cited by the Reviewer (95-80), the time was decreased compared with the previous one (110-50) because the second stage was carried out at a temperature (80oC) 30oC higher than the previous (50oC). The error bars were included in Figures 4 and 8, as requested by the reviewer (new pages 5 and 10).
- There is a lack of discussion in your study. I recommend splitting the "Results and Discussion" section into two distinct parts to ensure a comprehensive presentation and discussion of your findings.
R: Once the manuscript showed the results obtained in two different drying methodologies, we believe that a separation of the Results and Discussion into two distinct parts would become the reading and understanding of the information more difficult. Thus, we chose to keep in the revised form of the manuscript the discussion of the results closer to the place where they are presented, similar other articles published in the Molecules Journal.
- Finally, the entire manuscript should be presented in the English language to ensure universal readability and understanding.
R: The English language in the manuscript was improved, as suggested by reviewer.
I am confident that addressing these issues will significantly enhance your manuscript and better illustrate its contribution to the field.
Comments on the Quality of English Language: The manuscript contains several grammatical and writing errors which detract from its overall quality. For instance, the sentence, "The aspect of Spirulina samples dried by infrared radiation is shown in Figure 1", needs revision.
R: The English language in the manuscript was improved, as suggested by reviewer.
We would like to thank the reviewer for the time spent on reviewing our manuscript helping us improving the article.
Reviewer 3 Report
See attachment file

Some gramatical errors need to be corrected.
Author Response
Below we reproduce the reviewers’ original comments, followed by our responses (R) in italics. Changes in the text in response to the reviewers have been highlighted in revised manusript.
Reviewer #3:
Comments and suggestions for authors:
This is a good work that investigates the use of infrared and microwave radiation in the Spirulina platensis drying process. I have examined the submitted paper very carefully and I recommend its publication after some minor considerations. In the following, you can find my specific observations.
R: We thank the Reviewer for these positive comments
- Phenolic compounds have strong antioxidant activity and they processed bioactivities, including antioxidant activity. However, they are commonly vulnerable under thermal processing. These processes could affect their activity in the final product?
R: One of the objectives of this study was evaluate this vulnerability of the phenolic compounds in different drying conditions and with the use of continuous and intermittent drying. In addition to find the adequate conditions under which this effect in the antioxidant activity can be minimized.
- It is interesting the formation of darkening regions, probably formed for the oxidations process during the treatment. It is possible the polymerization process of polyphenols and because of that, the change in the material structure became more uniform and solid?
R: Different reasons are found in the literature to explain the formation of darkening regions, as the Mailard reaction and the oxidation during the drying, as related by the reviewer. We think that it is also possible that occurred the polymerization process of polyphenols. Thank you.
- If so, this material could be more resistant in different cocktion or solubilization methods. Did you test the final product to be ingested?
R: This work did not evaluate the ingestion of the final product or its use as food ingredient. It is also a good suggestion that we will consider in future works. Thank you.
- There are some grammatical errors that need to be corrected.
R: The English language in the manuscript was improved, as suggested by reviewer.
Comments on the Quality of English Language: Some gramatical errors need to be corrected.
R: The English language in the manuscript was improved, as suggested by reviewer.
The authors are grateful for the reviewer's observations and suggestions.
Related Papers Published in MDPI Journals
Abbaspour-Gilandeh, Y.; Kaveh, M.; Fatemi, H.; Khalife, E.; Witrowa-Rajchert, D.; Nowacka, M. Effect of Pretreatments on Convective and Infrared Drying Kinetics, Energy Consumption and Quality of Terebinth. Appl. Sci. 2021, 11, 7672. doi: 10.3390/app11167672
Tomas-Egea, J.A.; Traffano-Schiffo, M.V.; Castro-Giraldez, M.; Fito, P.J. Hot Air and Microwave Combined Drying of Potato Monitored by Infrared Thermography. Appl. Sci. 2021, 11, 1730. doi: 10.3390/app11041730
Lia Longodor, A.; Coroian, A.; Balta, I.; Taulescu, M.; Toma, C.; Sevastre, B.; Marchiș, Z.; Andronie, L.; Pop, I.; Matei, F.; Tamas-Krumpe, O.M.; Maris, S. Protective Effects of Dietary Supplement Spirulina (Spirulina platensis) against Toxically Impacts of Monosodium Glutamate in Blood and Behavior of Swiss mouse. Separations 2021, 8, 218. doi: 10.3390/separations8110218
R: These articles and new comments were included in the revised manuscript, as requested (new references 20, 24 and 31).
Round 2
Reviewer 1 Report
I thank the authors for their responses; they improved the content of the paper. I have no further comments and suggest publishing it in its current form.
Author Response
Comments and Suggestions for Authors
I thank the authors for their responses; they improved the content of the paper. I have no further comments and suggest publishing it in its current form.
R: The authors would like to thank the reviewer for the positive feedback on our manuscript
Reviewer 2 Report
The authors addressed my concerns.
Author Response
Comments and Suggestions for Authors
The authors addressed my concerns.
R: We thank the reviewer for the positive feedback